# AMDV Vaccine: Challenges and Perspectives

**DOI:** 10.3390/v13091833

**Published:** 2021-09-14

**Authors:** Nathan M. Markarian, Levon Abrahamyan

**Affiliations:** 1Faculty of Veterinary Medicine, Université de Montréal, Saint-Hyacinthe, QC J2S 2M2, Canada; nathan.marko.markarian@umontreal.ca; 2Swine and Poultry Infectious Diseases Research Center (CRIPA), Research Group on Infectious Diseases of Production Animals (GREMIP), Faculty of Veterinary Medicine, University of Montreal, Saint-Hyacinthe, QC J2S 2M2, Canada

**Keywords:** AMDV, Aleutian disease, mink parvovirus, Aleutian mink disease virus, vaccine

## Abstract

Aleutian mink disease virus (AMDV) is known to cause the most significant disease in the mink industry. It is globally widespread and manifested as a deadly plasmacytosis and hyperglobulinemia. So far, measures to control the viral spread have been limited to manual serological testing for AMDV-positive mink. Further, due to the persistent nature of this virus, attempts to eradicate Aleutian disease (AD) have largely failed. Therefore, effective strategies to control the viral spread are of crucial importance for wildlife protection. One potentially key tool in the fight against this disease is by the immunization of mink against AMDV. Throughout many years, several researchers have tried to develop AMDV vaccines and demonstrated varying degrees of protection in mink by those vaccines. Despite these attempts, there are currently no vaccines available against AMDV, allowing the continuation of the spread of Aleutian disease. Herein, we summarize previous AMDV immunization attempts in mink as well as other preventative measures with the purpose to shed light on future studies designing such a potentially crucial preventative tool against Aleutian disease.

## 1. Introduction

In the mid-1900s, some American mink farmers reported cases of a novel disease causing an enlargement of kidneys, the spleen and lymph nodes in the blue–gray variety of mink known as the Aleutian mink [1]. With time, the number of cases of this disease, called Aleutian disease (AD), rapidly increased throughout many ranches and the disease was eventually discovered to be due to a virus, as opposed to the initial false presumption of being a genetic disorder [2,3,4,5]. Today, AD is the most significant disease in the worldwide mink industry since it is responsible for causing infertility and the loss of animals, leading to low fur quality and, ultimately, significant financial losses for farmers [6]. The causative agent of this deadly and widespread disease is the Aleutian mink disease virus (AMDV), which is categorized as an *Amdoparvovirus* genus, one of the six genera of the *Parvovirinae* subfamily. This subfamily is part of the *Parvoviridae* family that belongs to the *Picovirales* order [7].

AMDV, being a parvovirus, is a single-stranded DNA virus possessing a small genome of around 4.8 kb in size with two large open reading frames (ORFs), among which are the left ORF at nucleotide positions 116–1975, the right ORF at nucleotide position 2241–4346, three smaller central ORFs and palindromic structures at both the 3′ and 5′ termini [3,8,9]. The left ORF encodes for three non-structural proteins (NS1, NS2 and NS3) that play a role in regulating gene expression and viral replication, whereas the right ORF codes for two viral capsid proteins (VP1 and VP2) serving as the key proteins for viral tropism and pathogenesis [10]. As for the palindromic structures at both 3′ and 5′ termini, these can fold into hairpin telomeres essential in the replication process [11]. Structurally, AMDV virions contain a predominantly negative strand DNA genome, measuring around 20 to 25 nm in diameter, are spherical, non-enveloped and are composed of 60 self-assembling proteins, including both VP1 and VP2 at a 1:9 ratio [11,12,13].

As for cell tropism, AMDV infects alveolar type II epithelial cells in mink kits and in macrophages for persisting infections, as well as B and T lymphocytes [14,15,16]. To enter macrophages and alveolar type II epithelial cells, AMDV has been reported to utilise antiviral antibodies using an Fc-receptor-mediated mechanism, which is known as an antibody-dependent enhancement of infection (ADE) [17,18]. In the case of other parvoviruses, many receptors are used to infect the target cells such as the transferrin receptor, bound by the canine and feline parvoviruses (CPV and FPV) and the erythrocyte P antigen for human parvovirus B19, whereas sialic acid is the main receptor for AMDV, minute virus of mice (MVM) and porcine parvovirus (PPV), respectively [19,20,21,22]. Upon receptor binding, parvoviruses are uptaken by clathrin-mediated endocytosis and slowly trafficked toward the cell nucleus, escaping the endosomal compartments with the help of the crucial phospholipase A_2_ (PLA2) motifs present on the VP1 capsid protein [23,24]. The PLA2s are enzymes that produce lysophospholipids and fatty acids from phospholipid substrates. Amdoparvoviruses, such as AMDV, on the other hand, are the only viruses in the *Parvovirinae* subfamily that do not contain this particular motif in the VP1 protein, and so, the endolysosomal trafficking remains to be clarified in future studies [24,25,26].

Upon the delivery of the single-stranded DNA genome of parvoviruses to the nucleus via the nuclear pore complex, the host cell DNA polymerase attaches itself to the 3′ terminal hairpin telomere of the genome to convert it into a double-stranded DNA [27]. During the S-phase of the host cellular cycle, from the P3 promoter, a precursor mRNA (pre-mRNA) is produced which is followed by polyadenylation and splicing to generate six distinct mRNAs named R1, R1′, R2, R2′, Rx and Rx′. From these, R1 and R1′ mRNAs encode for the NS1 protein; R2, which is the most abundant mRNA produced, encodes for the viral capsid proteins VP1, VP2 and the NSP2. The R2′ mRNA encodes for NS2, while the NS3 protein, whose function is not yet clear, is encoded from the Rx and Rx′ mRNAs [28,29,30,31]. In a permissive infection, the NS1 protein is cleaved by caspase proteins at two particular sites being 224INTD↓S228 and 282DQTD↓S286, which facilitates the localization of the full-length NS1 protein to the nucleus to initiate viral replication. This localization has been proposed to be possible by the oligomerization of cleaved NS1 protein with full-length NS1 [30,32,33]. In some cases, AMDV infection can be persistent where this caspase-mediated cleavage event of NS1 does not take place, resulting in a lower amount of viral DNA replicated and a limited amplification of mature virions [34,35]. NS1, with its helicase, DNA binding, ATPase and endonuclease domains, binds covalently to specific ACCA motifs at 5′ end of one of the strands of the double-stranded DNA genome and then performs a single-strand nick. This enables a process known as rolling hairpin replication (RHR) in which the terminal hairpin is synthesized and rearranged many times allowing the replication fork to be reversed and to generate many copies of the single-stranded DNA genome [36,37]. The latter is then bound by the VP1 structural protein when it enters the nucleus and, together with VP2, AMDV virions are then assembled to, subsequently, exit out of the nuclear pore complex and from the cell [20]. In a permissive infection, where the NS1 protein was cleaved prior to replication, the mature virions exit the cell by inducing cell lysis [38]. A proposed AMDV viral cycle is summarized in Figure 1.

At both the nucleotide and amino acid sequence levels, AMDV has an unusually high genetic variability where several strains have been identified each having a varying degree of severity of the disease [39,40]. In the case of the Utah 1, Ontario, Danish -K and United strains, these are known to be highly pathogenic, whereas SL3 and Pullman strains are moderately virulent and AMDV-G strain is non-pathogenic [41,42,43]. In newborn farmed mink (*Mustela vison*), Aleutian disease manifests as a fatal acute interstitial pneumonia with a mortality rate higher than 90% for mink infected with highly virulent strains and ranging from 30% to 50% in low-virulence strains (in infected newborn mink). The most serious form of the disease, however, is persistent and occurs in adults, where it is mediated by the immune system, causing plasma cell proliferation (plasmacytosis), an abnormal amount of gamma globulins present in the blood (hypergammaglobulinemia), the formation of infectious immune complexes and inflammation of kidneys (glomerulonephritis) [3,44,45,46]. AMDV is transmitted horizontally by air and contact between farms and also vertically from mother to offspring [46,47,48]. It has been shown that AMDV can also infect different species such as ferrets, otters, polecats, foxes, racoons, skunks, dogs, cats, and mice, but the disease has only been reported to be manifested in mink. Since this virus can infect other animals, the latter can be regarded as possible reservoirs facilitating AMDV spread [49,50,51,52,53].

Unfortunately, there are not any efficient methods to control the spread of this disease as AMDV is resistant to physical and chemical treatments, which is why farmers manually identify infected mink and kill those screened positive [18]. The diagnosis of AD mink takes place by looking for clinical signs and the detection of AMDV antibodies, which can be performed using non-specific methods such as iodine serum plate agglutination, where the excess of the gamma globulins levels in the serum can be visualized since certain gamma globulins can form complexes with iodine [54]. However, currently, more specific approaches are available such as counterimmunoelectrophoresis (CIEP), which involves using an electric current to assess binding resulting from the migration of unbound AMDV antigens towards the specific AMDV antibodies [55,56]. In recent times, this approach has been shown to be successful in reducing the prevalence of infected mink in Nova Scotia Canada, but has failed in eradicating AMDV from most of the infected farms [57]. Despite its advantages, CIEP is not a very sensitive diagnostic technique, can yield false negative results and is very time consuming [58,59]. In contrast, the enzyme-linked immunosorbent assay (ELISA) is another approach that is frequently used, which is based on detecting anti-AMDV antibodies in blood with the advantage of being automated and more sensitive [60]. Both CIEP and ELISA are very comparable and in good agreement, according to a study by Dam-Tuxen et al., where both techniques were evaluated in Danish mink [61]. Although being a more sensitive approach, ELISA has been shown to be a less accurate technique than CIEP, as reported by Farid and Rupasinghe (2016) [62]. Alternatively, the polymerase chain reaction (PCR) is another specific method that is used as a supplementary test, and this allows the detection of viral DNA by amplifying AMDV gene fragments [63]. A drawback of this method is that it does not detect all virus strains since AMDV has a great genetic diversity and is constantly mutating [64,65]. Another strategy to eliminate the harmful effects of AMDV is by selecting mink that are tolerant to this virus and this can be performed by identifying mink with a low antibody titer, which is performed in Europe and North America [58,66,67,68,69]. The selection for tolerance can also be performed by characterizing genomic regions of AD-tolerant mink [70]. Despite the different measures aimed to control the spread of AMDV and its harmful effects on mink, the virus is, unfortunately, still persistent on farms, which increases the demand of more effective AD preventative measures, as well as effective therapies.

Throughout the years, vaccines have been proven to be ingenious tools to be used against highly infectious pathogens and, thus, have been successful in eradicating diseases such as rinderpest in cows [71]. Consequently, one can assume that a plausible approach to combat against the persistent and highly consequential AMDV is to immunize mink against this virus. In fact, there were many attempts of researchers trying to develop an AMDV-vaccine using different approaches with a wide range of results obtained. For this reason, in this article, we review the several attempts of AMDV-vaccine design and the challenges faced in the process to provide a thorough understanding for future studies in the goal of eradicating this deadly disease.

## 2. AMDV Preventative Measures

Aleutian disease is one of the most threatening diseases in modern day mink farming. Further, domesticated mink escaping farms either by accident or with the help of animal activists pose a threat to wild populations of mink which could act as reservoirs of AMDV [72]. For this reason, the development of a novel AMDV vaccine is of crucial importance. In the meantime, it is of great interest to use alternative strategies to limit the viral spread and avoid huge financial losses to farmers and protect feral mink populations from potential spillovers of the virus. One strategy that is used for temporarily controlling the pathogen in farms is by the mandatory testing of mink to have an early detection and surveillance of AMDV. Common approaches of AMDV diagnosis consist in using serological testing techniques such as ELISA and counterimmunoelectrophoresis (CIEP) or PCR [73,74]. PCR is not as widely used as a routine serological testing method for herd screening due to a short-lived viral replication and viremia which causes negative PCR, while AMDV is sequestered in organs, making viral detection more challenging [44,58,75,76,77]. An additional reason for this is the cost of an essential DNA extraction step, which requires the clearance of residual inhibitors that would lead to the loss of sample DNA if not removed [78,79]. Additionally, in a recent study, it was shown that CIEP turned out to be more accurate than conventional PCR testing in black American mink infected with AMDV [75]. When taken together, early diagnostic approaches of AMDV-positive mink could provide reinforcing tools to help limit viral spread. Indeed, these have been somewhat effective in several countries, including Spain, where farms with AD have been reduced from 100% in 1980 to around 25% in 2019 [68].

Despite this encouraging statistic, however, the eradication of the disease remains challenging and has failed in many countries, including Canada, Denmark, and others [57,63,68,80]. This can be explained by the reinfection of farms due to viral persistence and by contaminated human individuals visiting farms, leading to a failure of eradication systems [81]. For this reason, some researchers explored the idea of immunomodulation in AMDV-infected mink. In fact, this approach has been investigated in a variety of animals, where the host immune system is modulated to be enhanced against infectious diseases. This can be achieved with numerous substances such as cytokines, microbial products, traditional medicinal plants, nutraceuticals and pharmaceuticals [82]. An example of immunomodulators are β-glucans, which are readily used in sheep and swine production systems [83]. For Aleutian mink disease, this strategy has also been explored by some researchers. This is the case of a study by Kowalczyk et al. [84] where female brown AMDV-positive mink were given methisoprinol (isoprinosine; inosiplex; inosine pranobex), a drug with immunopotentiating properties for which antiviral and immunostimulatory effects on Aujeszky’s and Newcastle disease have been reported by various groups [84]. Interestingly, the methisoprinol treatment of mink displayed a lower number of the AMDV DNA copies in the spleen and lymph node, as well as a higher fecundity compared to control mink [85]. More recently, Farid and Smith investigated the effect of kelp meal (*Ascophylum nodosum*) in mink infected with AMDV. From this study, it was shown that the supplementation of mink with kelp meal for 451 days did not improve the mink immune response to infection or virus replication. Interestingly, lower levels of blood, urea, nitrogen (BUN) and creatine were noticed, which suggested an improved kidney function [86].

Alternatively, another initiative for reducing the effects of AD on chronically infected farms is by passive immunity, which can provide immediate, but short-lived, protection to humans and animals [87]. In the context of Aleutian disease, the effect of the passive transfer of anti-ADV gamma globulins to newborn mink infected with a highly virulent strain of AMDV has already been investigated by Alexandersen et al. [88]. From their research, an acute AMDV infection was prevented by the passive antibodies, but mink still manifested a chronic infection [88]. The mechanism of the modulation from acute to chronic AMDV was then assessed in vitro, where infected alveolar type II cells treated with antibodies showed lower levels of AMDV replication and transcription intermediates. This was also noted in antibody-treated kits in vivo, which allowed the authors to suggest a role of antiviral antibodies in developing persistent infection in mink [88]. Furthermore, antiviral drugs are regarded as an important strategy to control the spread. Indeed, this strategy is very common and used for many infectious agents in humans and animals, such as human immunodeficiency virus (HIV) and Feline Herpes Virus (FHV) [89]. For AMDV, unfortunately, there is no approved antiviral treatments available for use; however, recently, Lu et al. [90] attempted a novel antiviral treatment using the “magnetic beads-based systemic evolution of ligands by exponential enrichment” (SELEX) strategy to generate aptamers for the AMDV VP2 protein. From their results, it was shown that, in vitro, the micromolar concentrations of the aptamers for AMDV VP2 protein were able to specifically inhibit by half the AMDV production by infected cells [90]. More investigation is, therefore, required in future studies in designing an adequate antiviral for AMDV.

Another strategy to prevent the harmful effects of AMDV is to select mink having low titers of anti-AMDV-protein antibodies in their blood, which is a common practice in many countries [58,66,67,68,69]. Besides this classical selection method, an alternative approach has been recently studied by Karimi et al. which is based on characterizing host-genomic patterns of AD-tolerant mink with the intention to select those that could be tolerant or resistant to AMDV. More specifically, the authors performed a sequencing approach to focus on genomic regions related to host specific immune responses. Using this strategy, two regions of the mink genome were shown to be strongly selected and these contained important genes involved in the immune response, viral–host interaction, reproduction and liver regeneration [70]. Additionally, since AMDV is a rapidly evolving pathogen, it is of great importance to be informed about current and past circulating strains to be able to develop epidemiological models, predictions and forecasting of the strains that may cause serious outbreaks. This knowledge would help us in devising an efficient and practical control strategy against this virus in the future, by setting up a database similar to the global initiative on sharing all influenza data (GISAID), which has been especially useful in the ongoing coronavirus disease-19 (COVID-19) pandemic in tracking novel variants [91]. Some studies have in fact performed phylogenetic analyses of the AMDV genomes deciphered by genetic sequencing. This was performed in order to develop new tools for outbreak investigation, the determination of virulence markers and development of more sensitive diagnostic tests in the future [92,93,94,95,96,97,98,99]. As an example of such work, Canuti et al. [93] studied the evolutionary dynamics of AMDV from 2004 to 2014 using sequences obtained from many regions of the world, including North America and Europe. From these, a very high viral genetic diversity as well as high rates of co-infection were noted in Newfoundland, Canada. Interestingly, a low level of diversifying selection was shown on structural proteins, which was explained to be due to the specific cell-entry mechanism of AMDV, which uses antibodies to enter cells. Finally, it was concluded that a great amount of circulating viruses on farms has the capacity to recombine and increase viral diversity by co-infections [93]. Therefore, summarizing, these methods could be of great use to control the spread of AMDV, while actively working on a development of effective vaccine candidates. A summary of these preventative measures is shown in Table 1.

## 3. AMDV Vaccine Attempts

### 3.1. Inactivated Vaccine

The use of inactivated and live attenuated virus-based vaccines dates back to more than a century. The principle of an inactivated viral vaccine consists of exposing a virus to chemical or physical agents, which disable its infectivity while allowing it to stimulate an immune response [100]. An example of this is by using formaldehyde, a cross-linking agent of amino acids, or, more modern approaches such as ascorbic acid or hydrogen peroxide, have been shown in recent years [101]. Currently, there are many inactivated viral vaccines available and approved for usage against viruses, including Hepatitis A, Japanese encephalitis virus, poliovirus, rabies and SARS-CoV-2 [102,103,104,105]. A study by Karstad et al., in 1963 [106], was one of the first attempts to develop an AMDV vaccine using this approach. These Canadian researchers used the inoculations of formalin-treated suspensions of tissues of AMDV-infected mink as a candidate vaccine [106]. It was shown that upon inoculating mink with this vaccine candidate, no plasmacytosis was noted in contrast to those inoculated with untreated diseased tissue. However, formalin-treated tissue failed to protect inoculated mink from the challenge with a virulent inoculum. Further, it was also noted that mink receiving three doses of the formalin-treated tissue suspension subcutaneously were not able to stimulate the immune system, i.e., the mink experienced plasmacytosis upon challenge [106]. Less than 10 years later, Porter et al. [107] tried a similar approach, where formaldehyde was used to inactivate AMDV-infected spleen and liver tissue homogenates. The inactivated homogenate was then administered to pastel mink and performed in parallel with the administration of a control group. From this, it was shown that there was no change in gamma globulin levels due to AMDV or the control group 33 days after inoculation. However, upon the challenge with a 10^2^ and 10^5^ tissue culture infective dose (TCID_50_), the gamma globulin levels in AMDV-vaccinated mink were slightly greater than the control group. Surprisingly, it was also reported that mink vaccinated against AMDV displayed more tissue lesions and enhanced plasmacytosis than the mink receiving the control vaccine, where both groups were challenged with the same dose of the virus. The authors then evaluated the effect of antibodies on this AD enhancement by giving 250 mg or 1 g of normal mink IgG antibodies to infected mink. A significantly enhanced disease was observed with acute necrotizing lesions. In contrast, this was not observed in infected mink receiving a lower dose of normal mink IgG, and in uninfected mink receiving both anti-AMDV IgG and normal mink IgG. Thus, it was suggested that antibody levels in the AMDV-vaccinated mink were not necessarily higher than in the control, since it was not noted as a significant number of lesions in the former compared to the latter. Moreover, based on immunofluorescence data, the authors suggested that observed enhanced pathology was due to antibody-complement-induced cytolysis followed by the complement-mediated attraction of polymorphonuclear leukocytes [107].

### 3.2. DNA Vaccine

Before AMDV was known to be the causative agent of Aleutian Disease, one study by Basrur and Karstad [108] investigated the effects of infection with DNA, extracted from the spleen tissue of mink with Aleutian disease. An in vitro inoculation of mink testis cells with viral DNA demonstrated a change in cellular morphology and abortive growth, which were absent in controls. This finding was also evaluated in vivo, on nine-month-old standard dark male mink, where plasmacytosis was observed in most of animals inoculated with the viral DNA. In contrast, the control group did not demonstrate such an effect, leading the authors to believe that the disease was of viral origin [108]. In more recent times, the use of DNA derived from viruses or other pathogens has been of great interest in the development of vaccines. This immunization method is referred to as DNA vaccination that consists of administering foreign viral protein-coding plasmid DNA into animals of interest [109,110]. Using the host cellular translation machinery, antigen proteins are then generated in situ followed by MHC-I and MHC-II pathways and the induction of CD8+ and CD4+ T cells, leading to an antigen-specific immunity [109,110,111]. DNA vaccines have been used in human clinical trials against infectious agents, for cancer immune therapies and for asthma and allergies as well as gene therapies for chronic diseases [112,113,114]. For animals, there are several DNA vaccines created against a variety of viruses. Some DNA vaccines have already been approved to be used, such as the first commercial DNA vaccine against H_5_N_1_ for chickens and the West Nile Virus vaccine for horses [115,116,117].

In the case of AMDV, there have been several attempts to develop a vaccine, but to date, none of them have been successful. One of those attempts was conducted by the group of Castelruiz et al. (2005), which designed a vaccine candidate based on an AMDV NS1 protein-coding plasmid (pNS1). This approach was first tested in vitro, where a high expression of the NS1 protein was revealed after transfecting Crandell feline kidney cells (CRFK) with the NS1 expressing plasmid. After confirming the NS1 protein expression in cell culture, the vaccine candidate was evaluated in vivo using three different groups of non-Aleutian female mink. In the first group, the animals were immunized with pNS1, followed by a booster inoculum containing recombinant NS1 protein (named “DNA + protein” group). The second group was identical to the first, with the exception of omitting the NS1-protein inoculation step (named “DNA only”). Finally, the third group received an empty plasmid, which did not contain the NS1 gene (named “control” group). After each of these groups were challenged, the “DNA + protein” vaccine cohort was shown to be the only group to exhibit antibodies against the NS1 protein at the time of challenge. With time, this was also observed in the other cohorts and, more specifically, at 12 weeks post infection, the “DNA only” group displayed a significantly higher anti-NS1 titer than the “control” group. A similar result was reported for all groups, when assessing anti-VP1/2 antibodies, with the only difference of not observing any VP1/2 antibodies on the day of challenge with AMDV. Interestingly, after 1 month of challenging mink with the virus, the “DNA+ protein” group exhibited higher levels of CD8+ T lymphocytes than the two other groups. This effect was suggested to be due to a memory response after an increase in interferon γ-producing lymphocytes for the same group was observed between 4 and 8 weeks post-challenge. A mild vaccine effect was suggested when noting that the “control” group displayed 10% higher gamma globulin levels than the other groups 8 weeks post-challenge. In terms of post-challenge deaths, most of the “control” group succumbed, where two of the latter were not likely to be AD related. In contrast, in the “DNA only” and “DNA + protein” groups, most of the animals survived. From the results of both AMDV NS1 DNA vaccinated groups, it was finally concluded that a mild vaccine effect was induced, but it was only partially protective, since there were still mink that died during the experiment [118]. Later, another attempt was undertaken by Liu et al. [119] to achieve the AMDV DNA-based vaccine (2018). For their vaccine candidate, the AMDV-DL125 strain was selected as a template since it displayed the highest TCID_50_ from virulent strains assessed in CRFK cells. From this strain, the whole genome was used to generate plasmid-vectored vaccines, by truncating regions of VP2 and NS1 genes using an overlap extension PCR. Similar to the study of Castelruiz et al., these plasmids with truncated regions were then tested in vitro on CRFK cells, where the successful infection of the cells with vaccine plasmids was confirmed by immunofluorescence, in contrast to controls, where no immunofluorescence signal was observed. Next, groups of six-month-old female mink were used to conduct an in vivo study. Most of these groups were administered with distinct DNA vaccine candidates, whereas the rest served as control groups, receiving phosphate-buffered saline (PBS), empty vector and AMDV-inactivated virus, respectively. Similar to Castelruiz et al., the anti-AMDV antibody levels in each group were shown to increase with time after challenge with AMDV. Interestingly, at the 24-week timeframe post-challenge, all groups demonstrated similar antibody levels in the serum, except for the inactivated AMDV-administered control group. The latter also displayed rapidly increasing levels of circulating immune complexes (CIC), suggesting that the inactivated AMDV vaccine could help accelerate the development of AD. For other groups, the ones inoculated with VP2 gene carrying plasmid with deletions at nucleotides coding for residues 428–446 and 487–501 (pcDNA3.1-ADV-428–487) had the lowest serum gamma globulin and CIC levels. This was suggested to be the most protective cohort; nonetheless, dead mink in each vaccinated and control group were observed with the earliest death occurring in the AMDV-inactivated group, followed by the others. By the end of the experiment (36 weeks post-challenge), the pcDNA3.1-ADV-428–487 group had the lowest mortality, and this allowed Liu et al. to suggest that this vaccine could be potentially used in mink populations in the future [119].

### 3.3. Subunit Based Vaccine

In 1981, the Food and Drug Administration (FDA) approved, for the first time, the plasma-derived hepatitis B virus (HBV) Heptavax-B vaccine (Merck) [120]. This was based on a heat-inactivated HBV surface envelope protein (HBsAg) isolated from the blood of asymptomatic HBV-infected patients and was shown to provide good protection in immunized individuals [121]. However, with the imminent HIV/AIDS global epidemic, concerns of infecting vaccinated individuals by blood with other pathogens limited this initiative, leading it to be discontinued in 1990 [122,123]. This encouraged the development of recombinant DNA technology, where the HBsAgs were produced in a yeast culture and this approach was used for the first time by the RECOMBIVAX vaccine (Merck Sharp and Dohme) approved in 1986 [123,124]. Protein subunit-based vaccines such as these primarily induce humoral immunity and exclude pathogens from their production, which reduces the risk of an incomplete activation, pre-existing immunity or causing disease [125,126]. Despite these advantages, the subunit vaccination, on its own, stimulates weaker immunological responses, which is why additional components known as adjuvants are added in vaccine doses [127,128]. Adjuvants, such as aluminium salts, can enhance cell-entry and preserve the structural integrity of the antigen, stimulate macrophages that promote helper T-cell responses, induce CD8+ cytotoxic T-lymphocytes as well as help in slowly releasing the antigens to prolong the exposure time to the immune system [129,130]. Currently, alongside HBV vaccination, the subunit-based approach is actively being tested for use in a variety of human pathogenic viruses such as Ebola and HIV [131,132,133].

In the case of Aleutian mink disease virus, this approach was investigated by Aasted et al. in 1998, where recombinant VP1/2 and NS1 proteins of AMDV-G were used to inoculate nine-month-old black female mink. Mink vaccinated with VP1/2 were challenged with AMDV which resulted in a higher death rate, more extreme hypergammaglobulinemia, higher NS1 and VP1/2 titers and higher counts of CD8-positive lymphocytes in lower peripheral blood compared to unvaccinated mink. The authors suggested that the observed outcome was caused by the antibody-dependent enhancement (ADE) of infection, a key mechanism in the pathophysiological changes of the AD, arguing that AMDV replicating cells are likely to be Fc-receptor positive. In the case of the group of mink vaccinated with a 10-fold higher dose of NS1, a more positive immunization effect was noticed. Throughout 11 months after challenge, lower levels of serum gamma globulin and CD8 positive lymphocytes were noted in addition to the lower death rates compared to a group of the non-vaccinated mink. The researchers did not detect vaccine-induced antibodies for VP1/2 and NS1 using ELISA assays. It was suggested that it could have been due to the induction of antibodies with a low affinity to native AMDV proteins since this higher affinity could have been developed to denatured the antigen protein vaccine present in antigen preparations prior to inoculation [134].

Considering the different vaccination approaches designed throughout the years against AMDV, the most successful attempt can be concluded to be the DNA vaccines, more particularly, those engineered by the group of Liu et al. [119]. Given the limited number of approaches for a successful vaccine design, more testing and the use of recent technological advancements in vaccinology can be employed to reinforce the fight against AMDV. Such methods are further elaborated in the Discussion section. The different vaccination attempts covered in this section against AMDV are summarized in Table 2.

## 4. Discussion

Throughout the years, many animal viruses have emerged and have caused serious financial and public health repercussions in many regions worldwide. A prime example of these is the Aleutian mink disease virus (AMDV), which poses a serious threat to mink since it is the most significant disease in their farming around the world [135]. Therefore, it is of great importance to produce effective treatments and vaccines to prevent significant losses caused by AMDV. As discussed in this review, there have been many attempts of producing vaccines against AMDV using different strategies. From these, three main techniques were used, including inactivated, subunit protein and DNA-based vaccines. In the case of inactivated vaccines, Karstad et al. showed an enhanced disease upon challenging mink that had been inoculated with formalin-treated AMDV-infected tissues, which was also shown by Porter et al. [106,107]. The latter additionally reported higher gamma globulin levels as well as more lesions in challenged vaccinated mink [107]. As for the design of subunit vaccines, Aasted et al. also demonstrated an enhanced disease after challenging VP1/2-vaccinated mink with higher death rates and extreme hyperglobulinemia [134]. In contrast, this effect was not noted in mink when a recombinant NS1 protein was used at a dose of 10-fold greater than that used for the VP1/2 inoculum. Instead, lower levels of gamma globulins and CD8 positive lymphocytes were present in the serum and less deaths were shown [134]. For DNA vaccine candidates, Castelruiz et al. obtained better results by using an AMDV NS1 coding plasmid either given alone or in combination with recombinant NS1 protein [118]. In fact, for both strategies, a mild vaccine effect as well as partial protection were found [118]. Similarly, a much promising result was reported by Liu et al., who designed seven plasmid-vectored vaccines, which were based on the genome of the infectious AMDV-DL125 strain [119]. From those, the best candidate showed partial protection and a higher efficacy based on lower levels of circulating immune complexes (CIC) gamma globulins in the serum [119]. This was based on the VP2 gene, which had deletions at nucleotides coding for residues 428–446 and 487–501 [119]. From all AMDV vaccination attempts, the most promising ones have been shown to be the DNA-based vaccines. Interestingly, for the other strategies, some authors have previously suggested an antibody-dependent enhancement of disease (ADE), which was used to explain the enhanced disease upon challenge of post-vaccinated mink [106,107,134]. Indeed, AMDV has already been reported to use the Fc-receptor-mediated mechanism to enter cells [17,18]. With this in hand, Bloom et al. investigated the molecular mechanism of ADE in the context of AMDV in vitro [18]. Namely, researchers evaluated effects of mono- and poly-clonal antibodies against short peptides designed from the immunoreactive VP2: 429–524 linear epitope using CRFK and K562_cells [18]. From their results, the VP2: 428–446 peptide-induced Fc-mediated ADE, the neutralization of AMDV and participated in an immune complex formation. A limited ADE was also noted in the case of the VP2: 487–501 peptide. It was finally concluded that this may be the mechanism by which capsid-based vaccines have the ability of inducing both neutralization and ADE, which agrees with the disease enhancement (vaccine-enhanced disease) shown in the inactivated and subunit protein approaches [18]. Further, this is also consistent with the study by Liu et al., where the exclusion of these sites led to a more efficient vaccine candidate [119]. Therefore, it might be suggested that in future vaccination attempts, sites from the immunoreactive VP2 segment should be excluded for a better response.

Another approach to this could be to analyze the full VP2 sequence to find epitopes that can further mediate ADE. One way to achieve this is by generating infectious clones of the virus and identifying capsid residues that are necessary for infecting cells in vitro. In fact, Xi et al. previously used this method to show that VP2 residues 92 and 94 are critical for AMDV replication in vitro [136].

Since there have been few vaccine technologies used to generate AMDV vaccine candidates, other vaccine development strategies or their combinations can be further used such as viral-vectored, virus-like particles and nanoparticle-based vaccines [137]. For example, for some diseases, viral vectors have been successfully used in combination with other vaccine approaches in a strategy called a heterologous prime-boost [138]. Lately, in the context of the current coronavirus-19 pandemic (COVID-19), one particular approach, based on the mRNA technologies, has been used in two licensed vaccines, which has helped significantly reduce infection numbers worldwide [139,140]. Both mRNA-1273 (Moderna) and BNT162b2 (Pfizer—BioNTech) rely on lipid-nanoparticle (LNP)-encapsulated mRNA expressing the pre-fusion stabilized spike glycoprotein, which is translated upon entering the cells [141,142]. With the great efficiency and success brought by these vaccines, a plausible approach could be to design a VP1/2 or an NS1-based mRNA vaccine for AMDV.

As the search for an efficient AMDV vaccine continues, novel methods are currently being explored to enhance vaccine (immunogen) delivery. This is the case of nanoparticle technology, which has been explored in the context of many viruses [143]. In fact, nano-vaccines (vaccines, where nanoparticles or nanomaterials are used as carriers) are created by displaying relevant antigenic sites on nanomaterials either by physical entrapment or by covalent binding [144]. Such vaccines have a similar size to the infectious pathogens, which makes it easier to enter the target cells, where they will be degraded and release the antigen over time [145]. In terms of their numerous advantages, these can induce immune responses, could be given intramuscularly or via mucosal sites, and they are stable at room temperature [144,146,147]. One interesting biodegradable nanovaccine approach that has been approved by the FDA is by using liposome-derived nanovesicles as carriers, since these can enhance the delivery of antigens to cells [148,149]. The use of such liposomal delivery systems has been previously reported to enhance immune responses against many viruses, including Newcastle disease virus [150,151]. Using this approach has also been shown to stimulate both Th1 and Th2 responses in response to an H3N2 influenza subunit vaccine candidate [152]. Another example of nano-vaccines is virus-like particles (VLPs), which mimic native viruses with the exception of not having the capacity to infect and replicate [153]. These are composed of self-assembling viral antigen proteins which can induce a stronger humoral immune response compared to single soluble antigens [154]. In recent years, there have been several approved vaccines using this technology, including Gardasil^®^ and Gardasil9^®^, against human papilloma virus (HPV), and many others [155].

Alternatively, another new vaccination method that is actively being explored involves the use of cell-derived bi-layered extracellular vesicles (EVs) such as exosomes, microvesicles and apoptotic bodies [156,157]. Indeed, these have great advantages in terms of delivery and can increase overall immunogenicity since they have immunostimulatory molecules on their surfaces such as MHC class I or II molecules [156,158]. Further, exosomes have the capacity to carry pathogen antigens, as was shown by Montaner-Tarbes et al. in a study analyzing exosomes from pigs infected with porcine reproductive and respiratory virus (PRRSV) [159]. From this, a specific reaction was demonstrated in a PRRSV RNA-negative and seropositive pig upon testing the exosome-derived viral proteins [159]. The microvesicle approach has also been tested by many researchers, including Rappazzo et al., where mice were vaccinated with microvesicles expressing ClyA surface protein fused with influenza matrix protein 2 [157,160]. Upon challenge with a virulent mouse-adapted H1N1 influenza strain, vaccinated mice demonstrated a full protection, and the passive transfer of their antibodies to non-vaccinated mice protected the latter when challenged [160]. Microvesical vaccines, such as liposomal nano-vaccine delivery methods, can also be given orally or nasally [161]. With the aforementioned advantages, these novel approaches for designing effective vaccines could be used to vaccinate mink against AMDV. This could be performed by incorporating key AMDV antigens in aerosol-based nanoparticles or EVs, which would help vaccinate a large number of mink rapidly from their cages by simply releasing these vaccines in the air. However, another important challenge faced, when designing AMDV vaccines, is that the virus has an unusually high genetic variability in both NS1 and VP2 proteins [39,40].

To enhance the design of future AMDV vaccines, one strategy can be the constant tracking of current and past circulating strains by phylogenetic analyses, which can help determine key virulence markers [92,93,94,95,96,97]. This is how Canuti et al. revealed that structural proteins were not under diversifying pressure in sequences analysed in Newfoundland, probably due to the conserved cell-entry mechanism of AMDV [93]. Moreover, in the wait of effective vaccines, there are many methods used and still being developed to control AMDV spread. A commonly used approach is detecting infected mink earlier on by serological testing techniques and by, subsequently, eliminating them [73,74]. Other techniques that have been explored or that are in development include the usage of passive anti-AMDV antibodies, antivirals, immunomodulators and the selection of mink with AMDV-tolerant traits [70,85,86,88,90]. Unfortunately, despite some positive results, these are not enough to eradicate this deadly disease, which is why the design of an effective vaccine is of primordial importance.

To conclude, AMDV poses a great threat to a huge number of mink either in farms or in in the wild and represents a potential risk to other species. In this review, the attempts to design a much-needed vaccine against this virus were covered, as well as the challenges that researchers are faced in designing a novel vaccine. For future studies, novel vaccine technologies should be aimed at, as well as trying to design better preventive measures to control viral spread.

## Figures and Tables

**Figure 1 viruses-13-01833-f001:**
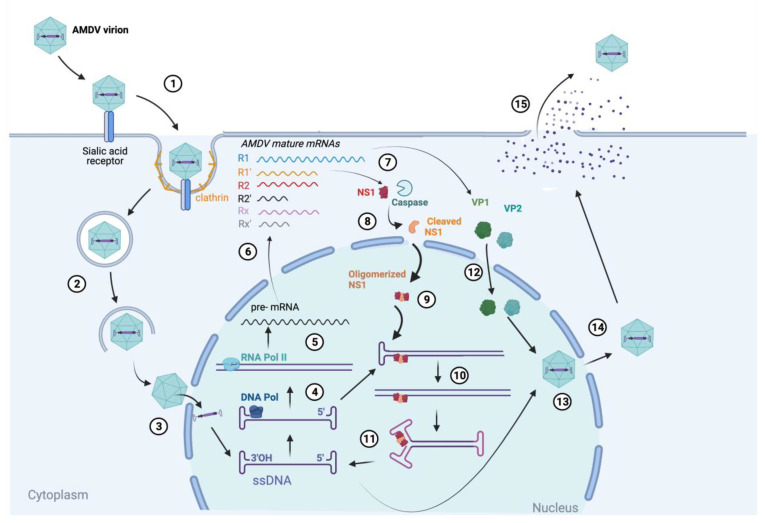
Proposed viral life cycle of Aleutian mink disease virus (AMDV). (1) The virus attaches and binds to the sialic acid receptor on host cells, triggering uptake by clathrin-mediated endocytosis. (2) Endolysosomal trafficking toward the nucleus. (3) Viral single-stranded DNA (ssDNA) genome is delivered into the nucleus. (4) Conversion of ssDNA into dsDNA by host cell DNA polymerase (DNA pol). (5) Transcription of double-stranded DNA (dsDNA) produces precursor mRNA (pre-mRNA) in the nucleus. (6) Splicing and polyadenylation of pre-mRNA yields six mature mRNAs, which exit the nucleus for (7) translation of viral proteins (only NS1, VP1 and VP2 shown). (8) Specifically, for AMDV: cleavage of NS1 protein by caspase proteins to facilitate localization of full-length NS1 to nucleus during permissive replication. (9) Full-length/cleaved NS1 oligomer enters the nucleus to covalently bind to the 5′ end of dsDNA, and (10) performs a nick. (11) Rolling hairpin replication (RHR) generates many copies of ssDNA genome. (12) Together, the VP1 and VP2 proteins enter the nucleus and (13) bind the ssDNA genome to form a mature AMDV virion. (14) Exit of AMDV virion from nucleus. (15) Release of AMDV virions through cell lysis. Created with BioRender.com (accessed on 15 August 2021).

**Table 1 viruses-13-01833-t001:** Some of the AMDV Preventative Measures Proposed: Diagnostic and Reduction Methods.

Measure	Results	Authors
Diagnostic method: CIEP ^a^	Reduction in the prevalence of infected mink in Nova Scotia Canada [57] and in Spain [68]; more accurate than ELISA [75]	Farid et al. (2012) [57], Pietro et al. (2020) [68], Farid and Hussain [75]
Diagnostic method: ELISA ^b^	Estimated sensitivity and specificity of 96.2% and 98.4%, respectively, for AMDV-VP2-recombinant antigen [60]; greater sensitivity than CIEP [61] but lower accuracy [61,62]	Knuuttila et al. (2014) [60], Dam Tuxen et al. (2014) [61], Farid and Rupasinghe (2016) [62] Chen et al. (2016) [74]
Diagnostic method: PCR ^c^	Relative diagnostic sensitivity of 94.7%, and relative diagnostic specificity was 97.9% [63]; specificity and sensitivity of 97.9% and 97.3%, respectively, for VP2 332–452 ELISA [74]; estimated specificity of 88.9% for AMDV-G NS1 probe-based real-time PCR [64]; lower sensitivity than CIEP [75]	Jensen et al. (2011) [63], Virtanen et al. [64], Farid and Hussain [75]
Immunomodulator molecules	Methisoprinol-administered mink showed lower number of AMDV DNA copies in spleen and lymph node and higher fecundity compared to control mink [85]; significantly lower levels of blood, urea, nitrogen and creatine in kelp meal-administered mink [86]	Kowalczyk et al. (2019) [85], Farid and Smith (2020) [86]
Passive Antibody Therapy	Prevention of acute AMDV infection by passive antibodies, but mink still manifested a chronic infection; reduction in mortality by 50 to 75% [88]	Alexandersen et al. (1989) [88]
Antiviral molecules	Specific inhibition of AMDV production in infected cells by AMDV VP2 aptamers: reduction of 47% supernatant concentration of AMDV compared to controls [90]	Lu et al. (2021) [90]

Diagnostic methods are used for detection, surveillance, for early detection and to control the spread. Immunomodulator molecules can be used to enhance the host immune system against infectious diseases. ^a^—counterimmunoelectrophoresis. ^b^—enzyme-linked immunosorbent assay. ^c^—polymerase chain reaction.

**Table 2 viruses-13-01833-t002:** AMDV Vaccination Attempts.

Vaccine Type	Approach	Disadvantage/Benefits	Authors
Inactivated virus	Formalin-treated infected kidney, liver and spleen suspension	No protection: challenged vaccinated mink developed plasmacytosis	Karstad et al. (1963) [106]
Inactivated virus	Formalin-treated infected spleen and liver tissue suspension	Enhancement of disease: challenged vaccinated mink displayed more tissue lesions/plasmacytosis than the non-vaccinated	Porter et al. (1972) [107]
DNA-based	NS1-coding plasmid	Partial Protection: majority of challenged vaccinated mink survived with the exception of a few deaths	Castelruiz et al. (2005) [118]
DNA-based	Whole gene-coding plasmid (pcDNA3.1-ADV) VP2-coding plasmid with deleted nucleotides coding for amino acids 428–446 (pcDNA3.1-ADV-428) VP2-coding plasmid with deleted nucleotides coding for amino acids 428–446 and 487–501 (pcDNA3.1-ADV-428–487) NS1-coding plasmid (pcDNA3.1-NS1) Truncated NS1-coding plasmid (pcDNA3.1-NS1-D) NS2-coding plasmid (pcDNA3.1-VP2) Truncated NS2-coding plasmid (pcDNA3.1-VP2-D)	Partial Protection: deaths observed in each category of challenged vaccinated mink, lowest number of deaths, the lowest serum gamma globulin and CIC levels for vaccinated cohort with pcDNA3.1-ADV-428–487	Liu et al. (2018) [119]
Subunit protein	VP1/2 and NS1 recombinant proteins ink aluminium hydroxide gel adjuvant	VP1/2—enhancement of disease: compared to control, a higher death rate and more extreme hypergammaglobulinemia in challenged vaccinated mink; NS1—partial protection: compared to control, lower death rates for challenged vaccinated mink	Aasted et al. (1998) [134]

## Data Availability

Not applicable.

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
