# Peer review of "AMDV Vaccine: Challenges and Perspectives"

_viruses, 2021, doi:10.3390/v13091833_

Round 1
Reviewer 1 Report
In this paper, the authors present a review based on Aleutian Mink Disease Virus (AMDV) description, and strategies to control viral spread, entitled “AMDV Vaccine: Challenges and Perspectives”.
The introduction is composed of a description of virus structure and life cycle (Figure 1), properly addressed the economic issues of the spread of Aleutian Disease (AD) and the need of efficient approaches to contain it. They address them following two main directions, namely “AMDV Vaccine attempts” and “AMDV preventative Measures”. The overall questions are well addressed, based on substantial amount of literature, but need some changes to raise a level of excellence.
First, as major comment, the title raised the question of AMDV vaccines as central, while preventative measures occupies quite the same space in the review. I suggest that the preventive measures paragraph could be shortened and displaced before the vaccines paragraph to emphasize and introduce the main subject of the paper.
The paper includes 3 illustrations (one figure and two tables) that need to be improved:
- Figure 1: Viral life cycle is simplified and could be applied to all Parvoviridae family. Specificity of AMDV should be properly notified, e.g. cleavage of NS1 protein by caspase (sustained by lines 73 to 78 in the text).
- Table 1: Description of each vaccination attempts should be more precisely described and main differences between references clearly noticed. Disadvantage/benefits column need further clarification: what mean partial protection? Reference (last column: authors) should use numbering to allow the reader to recover the proper bibliography.
- Table 2: Same remark for the last column (authors). To be instructive, table should contain the range of detection (for detection methods) and the level of decrease for reduction methods.
Minor comments:
- Line 35: a most recent report could be included, e.g. ICTV reports on Parvoviridae (2019)
- Line 56: “respectively” is not needed, as it refer to the same receptor
- Line 96: VP2 instead of Vp2
- References: DOI link conserved in references 24, 40, 57 and further.
Author Response
Dear Editor,
We would like to thank all three reviewers for their constructive feedback which was greatly useful in improving our manuscript. We have revised the manuscript keeping in mind all the suggestions made by reviewers. For each reviewer, we have given point by point responses, in which we explain how we have complied with all the suggestions. We truly hope that reviewers will appreciate our efforts on improving our manuscript. If there are further enhancements that can be done, we would be happy to receive additional comments by the reviewers for the next round of the revision. All in all, we hope that the reviewers will accept this updated version of our manuscript for the publication.
- Comments of Reviewer 1:
In this paper, the authors present a review based on Aleutian Mink Disease Virus (AMDV) description, and strategies to control viral spread, entitled “AMDV Vaccine: Challenges and Perspectives”.
The introduction is composed of a description of virus structure and life cycle (Figure 1), properly addressed the economic issues of the spread of Aleutian Disease (AD) and the need of efficient approaches to contain it. They address them following two main directions, namely “AMDV Vaccine attempts” and “AMDV preventative Measures”. The overall questions are well addressed, based on substantial amount of literature, but need some changes to raise a level of excellence.
First, as major comment, the title raised the question of AMDV vaccines as central, while preventative measures occupies quite the same space in the review. I suggest that the preventive measures paragraph could be shortened and displaced before the vaccines paragraph to emphasize and introduce the main subject of the paper.
Our reply: We appreciate the comment made by the first Reviewer. We agree with this suggestion and have brought the preventative measures paragraph before the vaccine paragraph. We have also removed diagnostic techniques that are not in common use such as the qPCR and the LAMP (lines 338-349 on original version). If Reviewer believes that information regarding the qPCR and LAPM is important to keep, please, let us know.
The paper includes 3 illustrations (one figure and two tables) that need to be improved:
- Figure 1: Viral life cycle is simplified and could be applied to all Parvoviridae family. Specificity of AMDV should be properly notified, e.g. cleavage of NS1 protein by caspase (sustained by lines 73 to 78 in the text).
Our reply: We appreciate the comment made by the first reviewer. We agree with this suggestion and therefore, we have rearranged the Figure 1 to incorporate the caspase cleavage and the proposed oligomerization for nuclear entry of full length NS1 protein by Best et al. (2003) . Here is the DOI to that paper: 10.1128/jvi.77.9.5305-5312.2003. PMID: 12692232; PMCID: PMC153974
- Table 1: Description of each vaccination attempts should be more precisely described and main differences between references clearly noticed. Disadvantage/benefits column need further clarification: what mean partial protection? Reference (last column: authors) should use numbering to allow the reader to recover the proper bibliography.
Our reply: We appreciate the comment made by the Reviewer 1. We agree with this suggestion and therefore, we have rearranged the Table 1 to add more clarification in the disadvantage/benefits column as well as the bibliography column. Since we brought the preventative measures before the vaccines paragraph, this table is now referred to as Table 2.
- Table 2: Same remark for the last column (authors). To be instructive, table should contain the range of detection (for detection methods) and the level of decrease for reduction methods.
Our reply: We appreciate the comment made by the Reviewer 1. We agree with this suggestion and therefore, we have rearranged the Table 2. We have outlined the main results discussed in one column and made to the bibliography column. Since we brought the preventative measures before the vaccines paragraph, this table is now referred to as Table 1. As the reviewer suggested, we decided to make the table focused on the detection and reduction methods and therefore, the “Selection for tolerant mink using genome-wide SNP markers” and “Phylogenetic tracking of AMDV strains for predicting outbreaks, epidemiological modeling” sections were removed from the table. This is why the new name of the table is “Some of the AMDV Preventative Measures Proposed: Diagnostic and Reduction Methods”.
Minor comments:
- Line 35: a most recent report could be included, e.g. ICTV reports on Parvoviridae (2019)
Our reply: Thank you. We agree and have added a more recent report (ICTV reports on Parvoviridae (2019) .
- Line 56: “respectively” is not needed, as it refer to the same receptor
Our reply: Thank you. We agree and have removed it.
- Line 96: VP2 instead of Vp2
Our reply: Thank you. We agree and have edited it (on line 99 of original version).
- References: DOI link conserved in references 24, 40, 57 and further.
Our reply: Thank you. We indeed removed the hyperlinks attached with the references.
Reviewer 2 Report
One comment regarding lines 438-440: Subjective and geographically specific to indicate that the wild mink populations are endangered. I would suggest deleting this part of the sentence.
Author Response
Dear Editor,
We would like to thank all three reviewers for their constructive feedback which was greatly useful in improving our manuscript. We have revised the manuscript keeping in mind all the suggestions made by reviewers. For each reviewer, we have given point by point responses, in which we explain how we have complied with all the suggestions. We truly hope that reviewers will appreciate our efforts on improving our manuscript. If there are further enhancements that can be done, we would be happy to receive additional comments by the reviewers for the next round of the revision. All in all, we hope that the reviewers will accept this updated version of our manuscript for the publication.
- Comment of Reviewer 2:
One comment regarding lines 438-440: Subjective and geographically specific to indicate that the wild mink populations are endangered. I would suggest deleting this part of the sentence.
Our reply: We appreciate the comment made by the Reviewer 2. We do agree with this suggestion and have deleted that sentence.

Reviewer 3 Report
This is a comprehensive Review addressing strategies pursued to inhibit spread of AMDV in mink farms. The manuscript is well organized and written and thus provides significant information to readers beyond the AMDV field.
There are, however, some minor points, which would be of interest being addressed:
- Since AMDV is a non-enveloped highly structured spherical particle it would be of interest, whether antibodies causing ADE only recognize linear or rather structural epitopes as well and whether these epitopes are at different surface areas as compared to neutralizing epitopes.
- Obviously there are AMDV strains differing in their intrinsic virulence that are recognized by differences in the respective capsids. It would certainly be of interest for the reader to know, whether changes in the respective capsids co-localize with ADE-peptides and/or neutralizing epitopes.
Author Response
Dear Editor,
We would like to thank all three reviewers for their constructive feedback which was greatly useful in improving our manuscript. We have revised the manuscript keeping in mind all the suggestions made by reviewers. For each reviewer, we have given point by point responses, in which we explain how we have complied with all the suggestions. We truly hope that reviewers will appreciate our efforts on improving our manuscript. If there are further enhancements that can be done, we would be happy to receive additional comments by the reviewers for the next round of the revision. All in all, we hope that the reviewers will accept this updated version of our manuscript for the publication.
- Comments of Reviewer 3
This is a comprehensive Review addressing strategies pursued to inhibit spread of AMDV in mink farms. The manuscript is well organized and written and thus provides significant information to readers beyond the AMDV field.
There are, however, some minor points, which would be of interest being addressed:
- Since AMDV is a non-enveloped highly structured spherical particle, it would be of interest, whether antibodies causing ADE only recognize linear or rather structural epitopes as well and whether these epitopes are at different surface areas as compared to neutralizing epitopes.
Our reply: We appreciate the comment made by the third reviewer. We do agree with this interest. As a matter of fact, we included an interesting paper in the Discussion (lines 658-662 in updated version) that showed that antibodies against the VP2:429-524 linear epitope are capable of inducing Fc-mediated ADE, neutralization of AMDV and participated in immune complex formation. It has not yet been shown if antibodies against structural epitopes are capable of causing ADE.
Here is the DOI for that paper (Bloom et al. 2021): doi:10.1128/JVI.75.22.11116-11127.2001
Obviously there are AMDV strains differing in their intrinsic virulence that are recognized by differences in the respective capsids. It would certainly be of interest for the reader to know, whether changes in the respective capsids co-localize with ADE-peptides and/or neutralizing epitopes.
Our reply: We appreciate the comment made by the Reviewer 3. We also do agree with this interest. However, as of now, we have not seen any papers showing whether changes in respective capsids of different AMDV strains co-localize with ADE-peptides/neutralizing epitopes. The main relevant paper is the one from (Bloom et al. 2021) which used AMDV-G to infect cells where different levels of ADE of infection were shown in presence of antibodies against different peptides, where VP2:429-524 linear epitope was capable of inducing Fc-mediated ADE. If we accidentally missed any report where this topic has been discussed, we would be happy to know that and add that information to our review.
